# Bisphenol A and Metabolites in Meat and Meat Products: Occurrence, Toxicity, and Recent Development in Analytical Methods

**DOI:** 10.3390/foods10040714

**Published:** 2021-03-27

**Authors:** Md Abu bakar Siddique, Sabine M. Harrison, Frank J. Monahan, Enda Cummins, Nigel P. Brunton

**Affiliations:** 1School of Agriculture and Food Science, University College Dublin, Belfield, Dublin 4, Ireland; md.siddique@ucd.ie (M.A.b.S.); sabine.harrison@ucd.ie (S.M.H.); frank.monahan@ucd.ie (F.J.M.); 2School of Biosystems and Food Engineering, Agriculture and Food Science Centre, University College Dublin, Belfield, Dublin 4, Ireland; enda.cummins@ucd.ie

**Keywords:** Bisphenol A, metabolites, meat and meat products, LC-MS/MS, GC-MS/MS

## Abstract

Bisphenol A (BPA) is a commonly used compound in many industries and has versatile applications in polycarbonate plastics and epoxy resins production. BPA is classified as endocrine-disrupting chemical which can hamper fetal development during pregnancy and may have long term negative health outcomes in humans. Dietary sources, main route of BPA exposure, can be contaminated by the migration of BPA into food during processing. The global regulatory framework for using this compound in food contact materials is currently not harmonized. This review aims to outline, survey, and critically evaluate BPA contamination in meat products, including level of BPA and/or metabolites present, exposure route, and recent advancements in the analytical procedures of these compounds from meat and meat products. The contribution of meat and meat products to the total dietary exposure of BPA ranges between 10 and 50% depending on the country and exposure scenario considered. From can lining materials of meat products, BPA migrates towards the solid phase resulting higher BPA concentration in solid phase than the liquid phase of the same can. The analytical procedure is comprised of meat sample pre-treatment, followed by cleaning with solid phase extraction (SPE), and chromatographic analysis. Considering several potential sources of BPA in industrial and home culinary practices, BPA can also accumulate in non-canned or raw meat products. Very few scientific studies have been conducted to identify the amount in raw meat products. Similarly, analysis of metabolites and identification of the origin of BPA contamination in meat products is still a challenge to overcome.

## 1. Introduction

Bisphenol A (BPA), 4-4’-(propane-2,2-diyl)-diphenol, is ubiquitously present in our water, environment, foods, and human specimens such as in blood and urine [1]. BPA is used in the formation of epoxy resin and polycarbonate, both of which have widespread industrial applications including in the production of digital media equipment (such as CDs and DVDs), medical devices, electronic equipment, sports safety equipment, cars, and baby bottles. Polycarbonate has also been used in the production of reusable bottles and food storage containers. In plastic manufacturing, BPA is normally used to enhance the transparency, durability, and impact strength of the final products [2]. In food packaging, epoxy resins are used as internal coatings of beverage cans to prevent direct contact of food contents with the metal surface. Food is considered the primary source of BPA exposure for human. Depending on the amount of exposure, several studies have indicated a relationship between BPA exposure and negative health outcomes [3]. Figure 1 represents the possible effects of BPA on human health.

BPA is rapidly absorbed and converts into conjugated forms namely BPA glucuronide and BPA sulfate in the gastrointestinal track and liver of humans and mammals [4]. The chemical structure of BPA and its metabolites are shown in Figure 2. The European Food Safety Authority (EFSA) has set the maximum amount of daily exposure of BPA in several food products at 4 µg per kg of body weight per day (µg/kg bW/day) [5]. However, according to the study conducted by Bemrah et al. some foods of animal origin have BPA levels of up to almost 400 µg/kg [6]. For example, liver samples and cooked veal contained BPA at levels of 395 and 224 µg/kg, respectively. In fact, the authors reported that meat and meat products, were amongst the seven food categories that exceeded threshold BPA concentrations, adding between 10 and 15 µg/kg BPA to the dietary intake. Interestingly, pork meat had BPA concentrations greater than 20 µg/kg [6]. 

However, reported BPA concentrations in meat products vary widely and include beef at 9–10 µg/kg [8], corned beef at 29–98 µg/kg [9], goulash at 9.6–22 µg/kg [10], and infant meat puree at 35.22 µg/kg [11]. It is worth mentioning that Sajiki et al. collected five meat samples from Tokyo metropolitan area and reported a lower BPA concentration (4 µg/kg) in chicken samples, while the highest amount of BPA was found in imported pork (20 µg/kg) [8]. Similarly, Thomson and Grounds [9] reported BPA levels in New Zealand corned beef was lower than the study conducted by Goodson et al. [12], where the origin of the samples was Brazil. Therefore, it is understandable that the variation of concentrations is related with the type of meat samples and largely with the type of can manufacturing materials. The estimated daily exposure of BPA through commonly consumed food products is detailed in Table 1.

In a more a recent study, Gorecki et al. measured BPA levels in foodstuffs collected from the French market between 2007 and 2009 [16]. The authors reported lower values than those outlined above of 51.3, 43.6 and 35.6 µg/kg as against values reported by Bemrah et al. of 395, 68.9 and 98.0 µg/kg in liver, roast pork and steamed salmon, respectively [6]. One plausible reason for this difference may be the variation in sample preparation and the sensitivity of analytical technique used. While Gorecki et al. analysed the collected samples individually in the raw state, Bemrah et al. used composite samples made from 15 sub-samples and analysed after culinary preparation [6,16]. 

Differences in the level of BPA exposure for different demographic groups such as pregnant women or children have also been reported. For instance, Bemrah reported the mean exposure to BPA for pregnant women to be 0.05 and 0.06 µg per kg body weight per day (µg/kg bW/day) for samples collected between 2007 and 2009 [6], while Gorecki reported exposures varying from 0.047 to 0.049 µg/kg bW/day in pregnant women for samples collected in 2015 [16]. Another study reported that the level of BPA exposure from canned foods for pregnant women from Southern Spain was 1.1 µg/kg bW/day [18]. On the other hand, Cao et al. analysed 158 food composites from Quebec City and found dietary intakes of BPA for infants were 0.17–0.33 µg/kg bW/day, for children aged from 1 to 19 years 0.082–0.23 µg/kg bW/day, and for adults 0.052–0.081 µg/kg bW/day [14]. Furthermore, the UK Food Standards Agency reported that BPA intake was 0.36–0.38 µg/kg bW/day for adults and 0.83–0.87 µg/kg bW/day for infants. However, this exposure level was calculated based only on the BPA concentration in canned foods in the United Kingdom [21]. For infants and young children under 36 months, BPA exposure increased with increasing age. This may be due to the introduction of common foods such as solid foods and beverages to the diet. However, for adults (18 years and over) BPA exposure decreased with increasing body weight which is likely due to dilution effect of body weight increases rather than decreases in BPA intake through decreased food consumption. Moreover, Thomson and Grounds reported that the mean and maximum exposure to BPA in New Zealand was 0.008 and 0.29 µg/kg bW/day, respectively, based on BPA levels in canned foods [9]. Although in most cases the daily exposure to BPA from dietary meat sources are claimed to be lower than the toxicological limit determined by the international authorities, the levels reported would have a significant contribution to the overall exposure. This is of concern considering the proven undesirable effects of even low-level exposures to BPA [22]. In addition, many of the exposure values are calculated on the basis of exposure from canned meat whereas the studies of Bemrah et al. and Gorecki et al. have demonstrated that a number of other types of meats and meat products may contain BPA at appreciable levels [6,16]. 

Therefore, more research is needed to address the contribution of different food categories (raw, processed and/or packaged) to the daily BPA exposure levels including their adverse health effects. BPA levels in meat and meat products, particularly in non-canned meats are rarely analysed. In addition, in the large studies published to date, there is a lack of understanding as to the impact of industrial processing and preparation, as well as household cooking procedures on BPA exposure. Therefore, the purpose of this review is to review and critically evaluate the available data on the regulatory aspects to control BPA exposure, occurrence of BPA and its metabolites in meat and meat products, possible routes of exposure in canned and raw products, as well as available procedures to detect and quantify BPA in foods, particularly meats.

## 2. Regulatory Aspects

### 2.1. Europe—Food Contact Materials

Commission Directive 2002/72/EC published in August 2002 authorized the use of BPA as a plastic monomer in the manufacturing of food contact materials [23]. The European framework regulation (EC) No 1935/2004 subsequently provided guidelines for any material coming into direct or indirect contact with food substances. It specifies that materials should be sufficiently inert so as not to lead to health-related issues in consumers, bring any unacceptable changes in the composition of foods or alter its organoleptic properties [24]. In 2006, EFSA stated that the highest level of BPA exposure in infants aged between 3 and 6 months old occurred via feeding from polycarbonate bottles, although the level of exposure was still below the tolerable daily intake (TDI) [25]. Despite EFSA’s statement of obtaining no harmful evidence of BPA for all groups of the population below the TDI, some questions regarding the possible toxicological relevance of BPA were raised. As a result, Commission Directive (2011/8/EU) amended directive 2002/72/EC relating to the use of BPA in food contacting materials. Specifically, Commission Directive 2011/8/EU, as a precautionary measure, restricted the use of BPA in the manufacturing of infant feeding bottles [26]. This amendment was subsequently added to Commission Regulation (EU) No 10/2011 for all plastic materials coming into contact with foods following the implementation of Commission Regulation (EU) No 321/2011. A further amendment to Regulation (EU) No 10/2011 was made regarding the use of BPA in coatings and varnishes coming into contact with foods. This change, reducing the specific migration limit for BPA from 60 to 50 μg BPA per kg of food, was published in Commission Regulation (EU) 2018/213 [27].

### 2.2. European BPA Limits in Foods

In 2006 the European Food Safety Authority (EFSA) completed its first full risk assessment on the use of BPA. As a result, EFSA recommended the TDI and reference dose for BPA to be set to 50 μg/kg bW)/day [25]. New scientific information was evaluated by EFSA in 2008, 2010, 2013, and 2015 resulting in a reduction of the TDI to 4 μg of BPA /kg bW/day in January 2015 [5]. Scientific experts have started to evaluate the recent toxicological data on BPA and the assessment has been scheduled for update in 2020 but has not yet been published. 

Apart from these EU-wide regulatory aspects, some national measures have also been adopted under the safeguard measures of EU framework regulations. For example, Denmark, France, Sweden, and Belgium have restricted the use of BPA in food packaging for young children. Denmark imposed a ban on the use of BPA for the manufacturing of plastic materials in contact with food intended for children aged 0–3 years [28]. In 2012, France adopted a law of suspending all BPA containing materials coming into contact with foods intended for children of 0–3 years of age from January 2013, and all other food contact materials from January 2015 [29].

### 2.3. USA

The Food and Drug Administration (FDA) approved the use of BPA under the food additive petition (FAP) process in 1960. In 2008, FDA released a draft describing the safety assessment of BPA using available data on the toxicity of BPA. In this draft, the FDA recommended that the No Observed Adverse Effect Level (NOAEL) for BPA for systemic toxicity be set at 5 mg/kg bW/day derived from two multigenerational rodent studies [30]. The 2009 CFSAN (Center for Food Safety and Applied Nutrition) low-dose updates along with 2011, 2012, 2014 Working Group assessments constituted a progressive series of evaluations with each subsequent memorandum building on the conclusions from the previous memoranda. While the FDA ruled that BPA-based epoxy resins would no longer be used as coatings in packaging of infant formula with effect from July 2013, this decision was taken largely due to abandonment by manufacturers, rather than due to concerns about BPA safety [31]. After finishing its ongoing safety review in 2014, the FDA announced that BPA was safe for the currently approved uses in food containers and packaging [32]. 

However, similar to Europe, several States within the USA have drafted legislation designed to offset the potential negative health effects of BPA used in many consumer products. For example, the Californian assembly bill 1319 (2011) prohibits the manufacture, sale or distribution of BPA-containing bottles or cups with a detection level above 0.1µg/kg if these products are meant to be filled with food items consumed by children, effective from July 2013. Similarly, the Connecticut House Bill 6572 (2009), Maryland House Bill 33 (2010) and Senate Bill 213 (2010), Wisconsin Senate Bill 271 (2010), Delaware Senate Bill 70 (2011), and Illinois Senate Bill 2950 (2011) prohibit the manufacture, sale or distribution of children’s food and beverage containers that contain BPA [33].

### 2.4. Canada

Canada became the first country to complete a comprehensive human and environmental risk assessment of BPA in 2008, and the Canadian health ministry proposed a regulation to prohibit the use of polycarbonate baby bottles in 2009 [34]. In 2010, Health Canada recommended that BPA be added to the toxic substances list under the Canadian Environmental Protection Act [35]. Following this and based on the overall weight of evidence, in 2012, Health Canada concluded that the current level of dietary exposure to BPA through food packaging uses was not expected to pose a health risk to the general population, including new-born and young children. Due to some uncertainties raised by some animal studies regarding the potential effects of low levels of BPA, the government expressed concerns for the safety of the products consumed by newborns and infants. It was, therefore, recommended that the general principle of ALARA (As Low As Reasonably Achievable) should be applied to continue efforts on limiting BPA exposure from food packaging applications for this segment of the population [36]. 

### 2.5. Rest of the World

The international regulation of BPA for using in food contact substances is not harmonized. However, several other countries also have legislation designed to either restrict or prohibit the use of BPA in food contact products, especially those designed and intended for use by young children. These include Argentina, Australia (voluntary phase out), Brazil, China, Ecuador, Japan, South Korea, and Turkey. 

## 3. Toxic Effects of BPA on Human Health

BPA has aroused consumer concern due to its potentially detrimental health effects. A review published by Almeida et al. have meticulously discussed the source and exposure criteria of BPA into canned foods and listed the negative health aspects as changes in neuronal development; changes in the reproductive system associated with decreased fertility; endocrine changes; metabolic, cardiovascular, and immunological diseases; damage to genetic material and cancer [37]. Indeed, a number of animal studies have found a relationship between BPA exposure and reproductive and developmental toxicity, immune toxicity [38], adverse neurological, endocrine, metabolic [39], cardiovascular effects and carcinogenicity [40].

Experimental studies have shown that BPA exposure may be a contributing factor in a variety of disorders affecting parts of the reproductive system including the hypothalamic-pituitary-ovary axis, ovaries, uterus, oviduct, and estrous cyclicity [41]. Animal studies have indicated that low and high doses (0.5 and 50 μg/kg bW/day) of BPA in neonatals inhibits the germ cell nest via altering expression of selected apoptotic factors [42]. BPA is also reported to alter oogenesis and follicle formation of rhesus monkey [43] and to affect the formation of primodial follicle by inhibiting meiotic progression in oocytes of pregnant mice [44]. Exposure to BPA is also linked to hyperandrogenism present in polycystic ovary syndrome in pregnant CD-1 mice [45].

Similar animal studies have also examined the effect of BPA exposure on uterus and uterine cells. For example, Vigezzi et al. reported the incidence of abnormalities in the luminal and glandular epithelium in mice with BPA exposure (50 µg/kg bW/day) [46], while uterine hyperplasia, stromal polyps, and retention of remnants of the Wolffian duct in the adult offspring of female murines (with BPA exposure 0.1, 1, 10, 100, or 1000 µg/kg bW/day) compared to the control has been reported by Newbold et al. [45,47]. The effects of BPA exposure on uterine function have been purported to be endomaterial cell proliferation, decreased uterine receptivity (100 mg BPA /kg bW/day) [48], and increased implantation failure (doses of 6.75 and 10.125 mg BPA/animal/day) [49]. 

As previously mentioned, animal studies have highlighted the impact of BPA exposure on the hypothalamic-pituitary-ovary axis. For example, low doses of BPA exposure (20 µg/kg bW/day) in adult female mice resulted in an increase in the level of Kiss1 mRNA [22]. This gene codes for the hormone kisseptin, which stimulates the secretion of gonadotropin-releasing-hormone (GnRH). The latter eventually stimulates the anterior pituitary to secret the gonadotrophic hormones follicle-stimulating hormone (FSH) and luteinizing hormone (LH) [50], which act on the ovary to support folliculogenesis. A dose-dependent (12–50 mg/kg bW/day) increase in the expression of levels of Kiss1 and GnRH observed in hypothalami of female and male pups was also demonstrated by Xi et al. [51]. Other experimental studies carried out on rodents have reported the effect of BPA exposure (from 25 ng to 100 mg/kg bW/day) on several reproductive parameters such as on vagina weights, egg shape, fertilization rate, distance between the genital pore and the anus of pore and the anus in newborns, time of vaginal opening and onset of the estrous cycle [52,53,54,55]. The resultant impacts have been listed as a decline in fertility and fecundity over time in female mice, morphological and functional alterations of the male and female genital tract and mammary glands, early opening of the vagina, and a significant decreased in the number of estrous cycle and days of estrus. 

It has been reported that, once in the body, BPA can disrupt the normal cell function by acting as an estrogen agonist [56], as well as androgen antagonist [57] which may affect the health of women during pregnancy. Investigations carried out on animals have shown that BPA exposure can impair the health through hampering the hypothalamas pituitary pathway. Animal studies varying the timing, length, and dose of BPA exposure during pregnancy have also been carried out and the results demonstrated that a high exposure to BPA (100 mg/kg bW/day) during pregnancy severely affects preimplantation embryonic development leading to the complete prevention of implantation in mice [48] as well as decreasing the number of live offspring rats [58].

A number of scientific works based on human and animal studies have investigated the long-term impact of BPA exposure on brain organization. Early exposure has been linked to several adverse outcomes such as reduced brain development and function [39] and alteration of brain sexual differentiation [59] in rats. Animal studies have also shown that different levels of BPA exposure can be the cause of behavioural changes such as aggression (2 or 20 ng/g bW/day) [60], anxiety (2 and 200 µg/kg bW/day) [61], cognitive deficits (100 µg/kg bW/day) [62], learning memory impairment (30 ng/g or 2 mg/g diet) [63] and modified socio-sexual behaviour (40 µg/kg bW/day) [64]. 

Furthermore, BPA can also contribute to the pathogenesis of cancer and several metabolic disorders. For instance, in vitro studies have shown that BPA (2.28 × 10^−9^–2.28 × 10^−3^ g/L) increased the proliferation of a human breast cancer line [65] and may also be a factor in the development of prostate cancer [66]. The latter research reported significantly higher concentrations of BPA (5.74 mg/g) in the urine of men with prostate cancer than the control group (1.43 mg/g). In addition, Menale et al. reported a link between BPA exposure and the modulation of glucose utilization in muscles, interference with adipose tissue endocrine function, neuroendocrine regulation of glucose metabolism and promotion of glucose metabolism dysfunction such as glucose intolerance and insulin resistance [67]. BPA exposure have also been reported to be linked to other metabolic dysfunctions such as obesity and type 2 diabetes [68] as well as cardiovascular toxicity [69] and immune toxicity [70].

Although there are numerous studies about the negative impact of BPA exposure, in 2015 EFSA announced that there are variables that make it impossible to claim that BPA is toxic to human health in respect to toxicokinetic differences between animal and human models, different routes of exposure, and the non-reproducibility of the studies [5]. Therefore, more scientific studies are needed in this area to have concrete evidence of the impact of BPA on human health.

## 4. BPA Contamination in Canned Meat Products

Research has shown that BPA can migrate from polycarbonate (PC) containing coatings to meat products contained in cans via two distinct routes, namely diffusion of residual BPA present in PC after the manufacture as well as hydrolysis of the coating polymer [70]. In general migration processes are also influenced by several factors such as food composition, type and duration of contact, temperature of contact, type of packaging material, nature, and amount of migrant compound [71]. Research by Geens et al. revealed that canned solid foods had 40 times higher BPA concentrations than the average BPA concentration found in canned beverages and about seven times higher than those observed in the liquid phase of the solid food canned [13]. The higher concentration of BPA in the solid foods such as meats compared liquid foods suggests the preferably migration of BPA from the coating to the solid portion of food with a higher fat content. The difference between the beverage and food could be related to the can types, coating conditions, sample matrix, presence of lipid, and sterilization differences between food and beverage cans [72]. A number of research articles have confirmed that almost 80 to 90% of BPA migration in food and beverage cans occurs during the sterilization step [8,73,74] which is typically performed at 121 °C for 90 min [73]. The latter authors also confirmed that the BPA concentration observed following the processing step did not change either following an extended period of storage (9 months) or damage to the can. Similarly, Stojanovic et al. examined the content of BPA in canned meat balls immediately after sterilization process and storing the samples at two different temperatures (20 and 40 °C) from 15 to 105 days. The authors observed the amount of BPA increased from 5 to 23.5 µg/kg (at 20 °C) and from 20 to 30 µg/kg (at 40 °C) after 15 days, further storage, however, lead to a very slow increase of BPA level [75]. Indeed, Munguia-Lopez et al. reported no significant differences in BPA concentration in samples stored for either 40 or 70 days, although BPA levels increased following storage for 160 days at 25 °C [74]. Sajiki reported that leaching BPA from epoxy resin in cans at 121 °C for 20 min was higher than that leached from cans held at or below 80 °C for 60 min [8]. 

Very few research studies have investigated the amount of BPA present in food stuffs with food packaging materials other than epoxy lined cans. In this regard, the experimental work of Geens et al. is interesting as the authors reported lower BPA concentrations (100 times lower) in beverages and foods packaged in packaging materials such as PET, glass and TetraPak compared to epoxy-lined cans [13]. Research carried out to understand the extent of BPA contamination in canned meat and meat products in samples from different countries such as Korea, China, USA, Japan, Canada, and New Zealand have reported a concentration range of BPA from ‘not detected’ to 98.3 µg/kg [17], <0.01–3.14 µg/kg [15], <0.01–8.78 µg/kg [20], 4–20 µg/kg [8], 1.2–35 µg/kg [76], and <20–98 µg/kg [9], respectively. This variation in the observed concentrations of BPA could be related to the different compositions of the meat samples, container materials, and food processing conditions investigated. Levels of BPA reported in several canned meat products are summarized in Table 2. Lately, it has also been speculated that the substitutes for and analogues compounds of BPA such as bisphenol S (BPS), bisphenol E (BPE) and bisphenol F (BPF) can also migrate from the cans into the foods. Since few studies have addressed this issue [15,76], more studies are needed to identify the source and concentration of these compounds in the meat and meat products.

## 5. BPA Levels in Raw/Non-Canned Meat Products

Although many studies have investigated the concentration of BPA in processed foods, BPA contamination of raw meats from the environment and/or contact with other BPA containing sources has been little investigated. In fact, EFSA published a scientific opinion stating that both canned and non-canned meat were the main BPA contributors among the different food items when it comes to BPA exposure. However, the concentration of BPA in the canned samples was found to be higher than in non-canned (raw) ones. Among the 19 different non-canned food categories, the highest BPA levels, on average, were found in meat products as well as seafood and fish products with levels of 9.4 and 7.4 µg/kg, respectively [5]. 

Reported occurrences of BPA in non-canned/raw meat and meat products are summarized in Table 3. Bemrah et al. reported that 17% of dietary exposure to BPA arose from non-canned animal products [6]. However, no specific explanation as to the source of the contamination in these foodstuffs was put forward. Gorecki et al. collected and analysed a broad range of 322 foods of animal origin representing two categories, namely pre-packed and cut-to-order, from several locations within the French territory [16]. These authors reported on average lower BPA concentrations in raw beef steak, pork chop, mutton, roast pork, and veal (2.93, 1.61, 3.19, 3.45, and 1.16 µg/kg, respectively), than the concentrations reported in the study of Bemrah (3.40, 16.95, 7.76, 12.44, 34.41 µg/kg, respectively) [6], while Adeyi and Babalola reported that BPA was not detected in raw beef, chicken, cheese, apple, tomatoes, beans and rice; and chicken eggs collected from Southwest Nigeria [77]. Furthermore, Gorecki et al. also investigated the concentration of conjugated, including monoglucuronide, diglucuronide, and sulfate forms of BPA in the collected samples [16]. However, no conjugated forms of BPA were found, suggesting that BPA was not metabolised from environmental intake by the animal but arose from contamination of the meat during post-mortem processing. According to the report published by the French Agency for Food, Environmental and Occupational Health and Safety (ANSES) on the average BPA contamination of foods, cut-to-order food items were statistically almost as likely to be contaminated as pre-packed foodstuffs. The results showed some evidence of contamination during processing. However, the potential contamination source was not identified due to a lack of precise data on the processing conditions such as sample handling, cutting location, or types of material used [81]. In this context, the authors want to highlight that the non-observing fact of BPA metabolites in raw meats could be related with the instability of these compounds particularly glucuronide (BPA-1G) under different preparation and storage conditions. For instance, Waechter et al. reported the rapid hydrolysis of BPA-1G into BPA aglycone in aqueous/organic solutions at lower pH (2) and 80 °C, the hydrolysis also occurred even at neutral pH and room temperature (22 °C) in some biological samples such as diluted urine and in rat placental or fetal tissue. Although BPA-1G was moderately stable in rat plasma up to 24 hr (at neutral pH and 22 °C), the authors have reported the degradation of metabolites present in urine/water mixture or urine/acetonitrile (25/75) stored at pH 9 at either 22 or 80 °C into unknown compounds [82]. Therefore, sample collection, storage, and preparation are crucial for the analysis of BPA metabolites in raw meat samples.

## 6. Recent Advancements in Analytical Methods

Taking into account present concerns with regard to BPA contamination of foods it is of paramount importance that sensitive, accurate, precise and robust methods are available for measurement of BPA and its metabolites. The selection of a robust method is the most important step in the detection and accurate quantification of the amount of BPA present in food samples. Analytical instrumentation is continuously evolving and becoming more sensitive, target-specific and robust and these developments have been applied to the separation and quantification of BPA in food and biological samples in a number of recent studies. The standard operating procedure for solid samples such as meats generally involves several steps including sample pre-treatment, extraction, clean-up and analysis. Figure 3 illustrates the steps typically involved in the determination of BPA and its metabolites in meat and meat products.

### 6.1. Sample Pre-Treatment

Sample pre-treatment is an important step for the isolation of targeted compounds from a bulk food matrix to avoid any interference in the final detection and quantification of analytes. Homogenization of meat samples (1–30 g) with subsequent removal of an aliquot is usually performed before the solvent extraction process. For meat samples, removal of fats and protein is preferable to avoid interferences in the final detection system. To achieve this, Shao et al. reported using celite and activated natural alumina [84], while Goodson et al. proposed the use of n-heptane to remove the fat from the samples [12]. However, the study conducted by Thomson reported that trimethylpentane was more effective for fat removal than *n*-heptane [9]. Although no loss of BPA individually for fat and protein removal step has been reported, addition of internal standard has been done to overcome potential loss of BPA during the overall sample preparation step. 

### 6.2. Extraction

Solvent extraction has commonly been used to extract BPA from solid food samples including meat and meat products. Acetonitrile is the most commonly used solvent during the extraction process [9]. However, some studies have also described the use of non-polar solvents such as *n*-hexane [10,85,86], *n*-heptane, or trimethylpentane in combination with acetonitrile for the extraction of BPA from fatty foods [87].

However, solvent extraction generally requires large volumes of solvent with long extraction times. In other applications, newly developed extraction techniques such as microwave assisted extraction (MAE), pressurized liquid extraction (PLE), matrix solid-phase dispersion (MSPD) including micro-extraction and quick, easy, cheap, effective, rugged, and safe (QuEChERS) extraction processes have recently been exploited to isolate target compounds from the sample matrices. Although these techniques can improve extraction efficiency, limited examples of their use for the extraction of BPA from meat and meat products can be found in the existing literature. Published extraction methods for bisphenols from meat and meat products are summarized in Table 4.

As an example, MAE has been used in the extraction of BPA from fish and seafood samples (prawns, crabs, cockles, white clams, and squids) using dichloromethane/methanol or tetramethylammonium hydroxide (TMOH) as solvents [90,91]. In addition, BPA extraction from fish liver [92] and meat products (pork, meat, rabbit, duck, and chicken) [84] has been performed using PLE and acetone/-n-hexane and dichloromethane as solvents, respectively. To optimize the extraction of BPA-diglycidyl ether (BADGE) from canned fish samples, Lapviboonsuk and Leepipatpiboon investigated the use of a QuEChERS method applying primary secondary amine (PSA) and C18 sorbents for the clean-up [93]. Results showed that the use of sorbents did not improve the quality of the extraction method, and that the QuECHERS method did not improve the recovery compared to other extraction methods used. Therefore, a simple sample preparation technique using acetonitrile with the addition of NaCl was proposed. Furthermore, Alabi, Caballero-Casero, and Rubio proposed the use of nanostructured liquids produced in colloidal solutions of amphiphilic compounds known as Supramolecular Solvents (SUPRASs) for improving the extraction of various types of bisphenol including BPA, BPB, BPF, BPE, bisphenol diglycidyl ethers as well as their derivatives from a wide range of food categories including vegetables, legumes, fruits, fish and seafood, meat product and grain [78]. The authors claimed that the mixed mode mechanism of the extraction improved extraction efficiency, while good sensitivity and selectivity was obtained by combining SUPRAS-based extraction with liquid chromatography (LC)-fluorescence detection. 

### 6.3. Clean-up

After the preliminary extraction, the crude extract typically requires further clean-up. In this regard, SPE has been frequently used for the removal of residual or co-extracted materials from the compound of interest, namely BPA. However, the performance of this method depends on the careful selection of a suitable sorbent material for the BPA analytes as well as selection of appropriate elution solvents. Isolation of BPA from homogenized samples of animal origin based on two consecutive SPE steps has been described by Bemrah and Deceunick [6,7,89]. These authors reported using a moderate/highly specific adsorption surfaced (>1000 m^2^/g) polystyrene-divinylbenzene polymer (PS-DVB) conditioned with methanol and water for the first stage, and a molecularly imprinted polymer (MIP) stationary phase conditioned with methanol, formic acid, acetonitrile, and water for the second SPE. While PS-DVB is able to retain hydrophobic compounds similar to reverse-phase chromatography, MIP stationary phases are based on a specific functionality to selectively separate estrogenic compounds due to phenolic interactions, thus making the retention of BPA more specific [94]. Shao et al. compared silica and amino-propyl SPE cartridges for the purification of crude meat samples and found recoveries of more than 91% for each cartridge [84]. However, samples cleaned with silica cartridges contained more fat after SPE. Other sorbents, such as an amino sorbent (Strata NH2 cartridge) conditioned with methanol/acetone and hexane were used by Liao and Kannan [15,20]. Sakhi et al. used a Florisil column conditioned with acetone/heptane (5:95, *v*/*v*) to remove fat from the extracts [83]. Recently, Cao et al. reported using a reversed-phase polymeric (Strata-X) SPE cartridge for the analysis of bisphenol S (BPS) including five other bisphenols from several food items including meat [76]. The mean recovery of the proposed SPE method ranged from 81 to 108% for all types of bisphenol present in food items. This SPE cartridge offers multiple retention characteristics including hydrophobic, π-π-interactions as well as hydrogen bonding. Moreover, as a result of its multimode interactions and its polymer base, it retains polar analytes more tightly than traditional C18 sorbents resulting in high recoveries and clean eluents [95]. 

### 6.4. Instrumental Analysis

A selection of instrumental analysis of BPA from meat and meat products are listed in Table 5. The cleaned-up extract is used in the final step for analysis by either liquid chromatography (LC) or gas chromatography (GC). A variety of detection methods have been used in LC including ultraviolet (UV), fluorescence, and mass spectrometry (MS). In recent times, fluorescence and MS-based detectors are more frequently used because of their greater specificity as well as higher sensitivity [96]. The use of a traditional C18 column with 3.5–5 µm particles and 4.6 mm internal diameter has been frequently reported for the HPLC analysis of BPA from food materials [78,97,98]. However, C18 columns with smaller particle sizes such as 1.7 um have also been used for the determination of BPA from canned fish, vegetables and sauces [99]. Gallart-Ayala, Nunez and Lucci reported that columns with smaller particle size (sub-2 µm) improved chromatographic resolution and decreased analysis time [98]. While the traditional C18 column was the most popular choice (67%) [4], other stationary phases such as PentaFluoroPhenyl (PFP) bonded phase column have gained acceptance due to the alternative selectivity they provide. In comparison to the traditional alkyl phases, PFP bonded phases have been shown to provide enhanced dipole, π-π, charge transfer, and ion-exchange interactions [100]. For instance, Battal et al. reported enhanced selectivity for a PFP column (100 × 3.0 mm, 3 μm particle size) over the C18 liquid chromatography column (100 × 2mm i.d., 3.5 μm particle size) and C8 liquid chromatography column (250 × 4.6mm i.d., 5 μm) [101]. Similarly, C18 columns containing superficially porous particles (SPP) are gaining popularity over the frequently used fully porous particles (FPP). Gallart-Ayala, Moyano, and Galceran reported achieving efficient chromatographic separation of bisphenols (BPA, BPF, BPE, BPB, BPS) with a rapid analysis time (less than 3 min) by using a SPP C18 column [102]. Better chromatographic efficiency through the use of a SPP C18 column as compared to a FPP column for the analysis of BPA from fruit drinks has been also reported [103]. GC analysis, on the other hand, typically deploys a (5%-phenyl)-methylpolysiloxanecapillary column such as DB-5 or HP-5 with a length of 30 m, an internal diameter of 0.25mm and film thickness of 0.25 µm [9,14]

#### 6.4.1. HPLC-UV

The UV absorption wavelength of BPA has been reported to be 227–230 nm [97]. However, BPA also absorbs at a wavelength between 272 to 280 nm [105]. Cao, Zhuang, and Liu also reported that the maximum UV absorption of BPA was at 278 nm [106]. Similarly, Watabe et al. developed a HPLC method for the detection of BPA from lake water using a UV wavelength of 275 nm [107]. Previously, Takino et al. proposed a method using HPLC system with UV detector for the determination of BPA in canned fish and meat samples [104]. These authors used a C18 column (4.6 mm i.d. × 150 mm) and a 60% methanol mobile phase at a flow rate 0.8 mL/min, thereby achieving a limit of detection limit (LOD) of BPA of 25 ng/g. Furthermore, Aristiawan et al. calculated an LOD and limit of quantification (LOQ) of 0.8 mg/kg and 1.5 mg/kg, respectively, while the recovery was 89.4% for the tuna fish samples using HPLC-UV [97].

#### 6.4.2. HPLC-FLD

Grumetto reported that HPLC-FLD (fluorescence detector) resulted in less chromatographic interferences and improved sensitivity than a HPLC-UV method [108]. FLDs are recognized for high sensitivity, selectivity and repeatability. BPA shows two excitation maxima at 212–226 nm and 272–278 nm and one emission maximum located between 297 and 308 nm depending on the solvent used (methanol, ethanol, diethylether, or water). The maximum fluorescence signal is obtained in polar organic solvents such as methanol or ethanol with a marked decrease being observed in water [109]. Braunrath et al. used HPLC–FLD both in isocratic and gradient modes for the determination of BPA in a wide range of canned food stuffs including meat goulash [10]. For each HPLC calibration curve, good linearity (*r*^2^ > 0.9993) for the standard solution from 0.2 to 50 ng/mL was obtained. LOD was reported to be between 0.2 and 0.8 ng/mL using an isocratic mobile phase depending on the beverage samples, while 0.4 and 0.5 ng/mL for gradient program with goulash and fish, respectively. Similarly, Bendito, Bravo, Reyes, and Prieto used a fluorescence detection based HPLC method with a gradient elution program for the quantitation of BPA in different food including canned tuna, mackerel, meatballs, and lean pork [79]. Their LOQ was reported to be dependent on the amount of sample used and ranged from 29 to 15 ng/g when either 200 or 400 mg of sample was used. In addition, their method achieved recoveries of between 90 and 99% while detecting BPA levels in fish and meat samples of between 20 and 129 ng/g and not detected and 37 ng/g, respectively.

#### 6.4.3. HPLC-MS

Although fluorescence detection is suitable for a wide range of food samples, researchers often suggested to use MS-based techniques to confirm BPA in complex food matrices and to reduce errors in quantification [18,110]. Therefore, systems such as LC-MS or GC-MS are commonly used for BPA quantification. In MS methods, internal standards such as isotope labelled ^13^C-BPA [13] and deuterated BPAd_16_ [88,111] has been frequently used to overcome the loss of analytes during sample preparation. In this regard, LC-MS and LC-MS/MS demonstrate excellent sensitivity and precision for the analysis of BPA in food, environmental and biological samples. In most cases, negative ion electrospray ionization (ESI) has been used. Compared to LC-MS, LC-MS/MS is more specific and selective for BPA as it allows single or multiple reaction monitoring (SRM or MRM) thus giving more confidence in peak identification as compared to LC-MS [8,96]. The study conducted by Sajiki et al. showed that the LOD of BPA by LC-MS and LC-MS/MS was 0.1 ng/mL for both methods, with LC-MS/MS being more precise (%RSD = 1.2) than LC-MS (%RSD = 3.2) [8]. LOQ was reported as 1 µg/kg for the determination of BPA in meat samples by combining accelerated solvent extraction with LC-MS/MS [84]. Moreover, the UHPLC-MS/MS method for the detection of conjugated metabolites of BPA and BPS in foods of animal origin described by Deceuninck et al. showed LODs in muscle samples of 0.02, 0.09, 0.04, 0.12, and 0.50 µg/kg for the metabolites BPA-monoglucuronide (BPA-1G), BPA-diglucuronide (BPA-2G), BPA mono-sulfate (BPA-1S), BPA-disulfate (BPA-2S) and BPS-monoglucuronide (BPS-1G), respectively [7]. Finally, the average recovery of BPA from meat samples was reported to range from 91% to 108% when using LC-MS/MS [76,84].

#### 6.4.4. GC-MS

GC-MS systems are also frequently used for the quantification of BPA. Helium is usually employed as a carrier gas at a constant flow ranged from 0.76 mL min^−1^ [9] to 1 ml min^-1^ [81]. Spitless injection mode has been applied, while the injection temperature was reported as 280–300 °C. The oven temperature was reported to start from 100–150 °C, and to reach 280–300 °C. The MS was mostly operated in electron ionization mode (70 eV) [88,89,111], however, MS operated in negative chemical ionization also has been reported [112]. Ion source temperature and MS quadrupole temperature was used at 230 and 150 °C, respectively [81,88,89,111]. Before analysing samples by GC-MS, a derivatization step is needed to modify the slightly polar properties of BPA molecules thus producing a compound that is suitable for GC analysis (lower polarity, higher volatility, higher stability, and higher peak efficiency and detectability) [113]. Frequently used derivatization approaches include silylation [114], and acetylation [12]. Extracts obtained after clean-up steps were derivatized with commercially available derivatization reagents, such as acetic anhydride [9,12,14], N,O-bis(trimethylsilyl)trifluoroacetamide (BSTFA) [115], N-methyl-N(trimethylsilyl)-trifluoroacetamide [6], and pentafluorobenzoylchloride (PFBCl, 5% *v*/*v* in hexane) [13].

The mass spectrum of extracts derivatized with acetic anhydride displayed m/z 213 as the most abundant ion [9]. This is probably due to benzylic carbocation, resulting in the loss of methyl group from BPA. The resulting ion with m/z 213 was very stable resulting in it being the base peak in the mass spectrum [28]. Munguia–Lopez et al. reported m/z 213 as the main ion with further fragments including m/z 228 (molecular ion), 110, and 91 [115]. While m/z 213 ion was used for the quantification, m/z 228, 255, 270, and 312 have been used for the qualification of BPA in extracts from meat and fish samples in GC-MS [81,88]. On the other hand, mass spectrum of BPA derivatized with BSTFA containing 1% trimethylchlorosilane (TMCS) showed m/z 357 as the most abundant ion, used for the quantification, with m/z 372, 207, and 73 being used for the confirmation [114,116]. Geens et al. used selected ion monitoring mode for the analysis of BPA derivatized with PFBCl where the quantification ion was m/z 616, with m/z 408 being used for confirmation [13]. However, Jurek and Leitner reported m/z 601 and 616 as quantifier ions for GC-MS using electron ionization and negative chemical ionisation (Isobutane was used as ionization gas), respectively, while m/z 195 and 617 were used as qualifier ions [112]. Finally, the fragmentation pathway of BPA was studied using Orbitrap MS in negative ion mode. Product ions at m/z 211.07618, 133.06488, and 93.03336 were detected using this method probably due to the losses of CH4, phenol (C6H6O), and isopropenylphenol (C9H10O) from the precursor ion [117].

Goodson et al. reported the LOD and recovery of BPA from canned foods using GC-MS as 2 µg/kg and 81–103%, respectively [12]. On the other hand, Thomson and Grounds reported a LOQ of <10 µg/kg for foods of low-fat content (<1%) and <20 µg/kg for foods containing >1% fat [9]. In comparison to the reported values of BPA detected using LC-MS (LOD: 0.1–1.2 µg/kg, LOQ: 0.01–3.14 µg/kg, and recovery: 62–120%), meat samples analysed with GC-MS showed LOD: 0.00013–1 µg/kg, 0.0004–20 µg/kg, and recovery 42–112% (data reported in Table 5). Although acceptable ranges of LOD and LOQ of BPA from food samples has been mentioned in the literature, GC-MS is less used than LC because of the additional sample preparation associated with the derivatization step.

## 7. Conclusions

BPA exposure can trigger several negative outcomes for human health especially in pregnant women, infants, and children. Among the different sources, diet is considered the main route of inclusion of BPA into the human body. Findings from this literature review indicate that the inclusion of canned meat products are the substantial contributors to total BPA exposure. This is because of the migration of BPA from the can lining materials into the solid parts of the meats during processing. Therefore, food and health regulatory associations of many countries have addressed this concern by either strictly banning the use of BPA or by only allowing certain food contact materials. The migration process can be influenced by several factors such as the packaging materials used, food composition, contact duration as well as temperature of processing. Very little research has been carried out on the contribution of BPA from other commonly used meat packaging materials such plastic containers, wraps, bags, and films. It appears that BPA can also accumulate in raw meat from environmental sources due to its ubiquitous presence in the environment. Among non-canned foods, meat and fish products have been found to contain the highest level of BPA contamination. 

Accurate measurement of BPA concentration in foodstuffs is necessary to assess the health risk associated with exposure. Sample preparation is considered to be a crucial step before the determination of analytes through chromatographic analysis of free BPA and metabolites. However, few studies have been conducted to confirm the presence or absence of metabolites in meat samples and therefore the source of contamination especially in raw meats is difficult to discern. To date, results have shown no detectable amount of conjugated BPA indicating that the contamination mostly occurred from yet unknown sources on industrial production lines. Data on the contribution of household practices such as the use of plastics to store foods, polycarbonate cooking utensils, and heating of foods to levels of BPA contamination in foods is limited. Therefore, further research is required to carefully determine the amount of BPA, both in free and conjugated forms, present in meat samples to identify the contamination route and thus formulate practices and recommendations to reduce the amount of contamination to meet consumer safety requirements.

## Figures and Tables

**Figure 1 foods-10-00714-f001:**
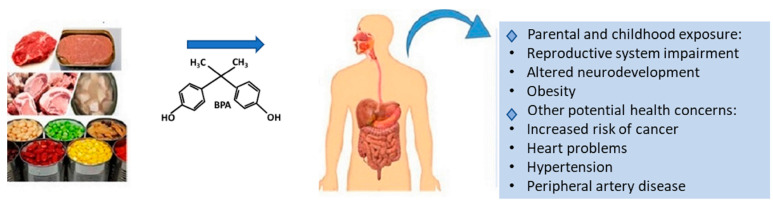
Health issues related with the intake of Bisphenol A from dietary sources.

**Figure 2 foods-10-00714-f002:**
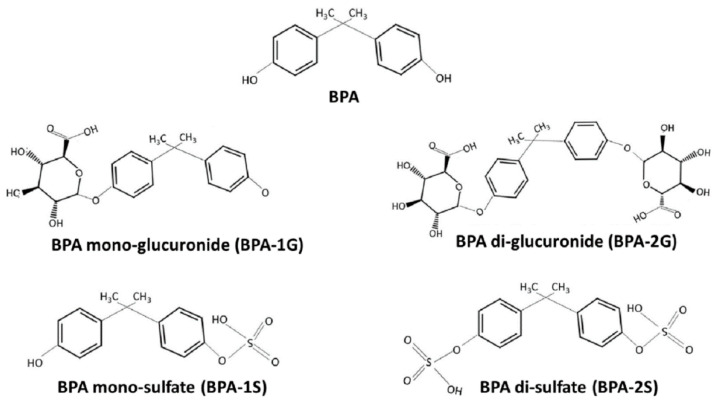
Chemical structure of BPA and its metabolites (Adapted with permission from [7]). Copyright 2019 Elsevier.

**Figure 3 foods-10-00714-f003:**
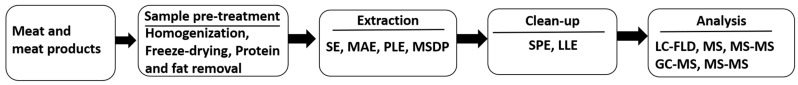
Flowchart for the determination of BPA/BPA metabolites from meat samples. SE: Solid extraction, MAE: Microwaved assisted extraction, PLE: Pressurized Liquid extraction, MSDP: Matrix solid-phase extraction, SPE: Solid phase extraction, LLE: Liquid-liquid extraction, LC-FLD: Liquid chromatography fluorescence detector, MS: Mass spectrometry, MS-MS: Chromatography with tandem spectrometry, GC: Gas chromatography.

**Table 1 foods-10-00714-t001:** Estimated daily exposure of BPA from commonly consumed foods.

Country	Selected Food/Food Groups Analysed	Population Groups	Dietary Exposure of BPA(µg/kg bW/Day)		Reference
			Mean ± Sd *	Range (Min–Max)	
Belgium	Canned beverages and foods	Adults	0.015	-	[13]
Canada (Quebec city)	Dairy, meat, fish, soup, bread and cereal, vegetable, fruit, beverages, baby food and fast food	Infants Children (1–19 years) Adults	-	0.17–0.33 0.082–0.23 0.052–0.081	[14]
China	Cereal products, meat and meat products, fish and seafood, dairy products, bean products, vegetables, snacks, and beverages	Adult men Adult women	0.484 0.494	-	[15]
France	Bread and cereals, dairy and egg products, meat, poultry and game, fish and seafood, fruits and vegetables, beverages, and fast foods	Infants Children and adolescents Adults Pregnant women	-	0.12–0.14 0.05–0.06 0.038–0.040 0.05–0.06	[6]
France	non-canned foods from animal origin	Children and adolescents Adults Pregnant women	-	0.048–0.050 0.034–0.035 0.047–0.049	[16]
Korea	Vegetables, fruits, fish, meat, tea, and coffee (canned)	Adults	1.509	-	[17]
New Zealand	Fruits and vegetables, fish, soup and sauces, canned meat, spaghetti and baked beans, infant foods, and beverages	Adult (60 kg) Adult (75 kg)	0.078 0.063	-	[9]
Spain (Southern)	Fish, meat, vegetables, pulses, and soft drinks (canned and microwave containers)	Pregnant women	1.1 ± 0.84	-	[18]
Sweden	Cereal products, fish, dairy and products, fruits and vegetables, and beverages	Adults (17–79 years)	-	0.04–0.07	[19]
United states	Solid foods, oil, beverages, and dairy products	Toddlers Infants Children Teenagers Adults	0.243 0.142 0.117 0.0636 0.0586	-	[20]

BPA: Bisphenol A, * Sd: Standard deviation.

**Table 2 foods-10-00714-t002:** Level of BPA found in canned meat and meat products.

Figure	Sample Description	Concentration of BPA (µg/kg)		Reference
		Mean ± Sd *	Range (Min–Max)	
Beef Chicken	Three different samples were collected for each category at three different local markets.	12.7 ± 7.7 4.42 ± 1.5	5.88–21.3 2.94–6.36	[77]
Meat balls Tripe	Samples were collected from local markets and kept at room temperatures before opening.	82 ± 3 62 ± 2	-	[78]
Lean pork	Lean pork cooked in its own juice. Cans stored at room temperature.	37 ± 5	_	[79]
Goulash	N/A	27 ± 4	9.6–22.0	[10]
Luncheon meats Meat soup	Foods were prepared and combined into food composites according to established procedures.	10.5 29.1	-	[14]
Canned meat Infant meat puree	N/A	19.39 35.22	-	[11]
Sausages	Samples were randomly chosen from local supermarket. Stored at room temperature and analyzed within seven days after purchase.	26.7	-	[13]
Hot dogs Chopped pork and ham Corned beef	Three cans of each samples were purchased from retail outlets. Collected samples were stored at room temperature.	-	21–33 16–17 59–70	[12]
Minced Beef	Empty cans were filled with foods processed at 121 °C for 90 min, sealed, and either stored at 5 and 20 °C for up to 9 months or at 40 °C for 3 months	53.8 ± 7.6	-	[73]
Cooked pork (Spam) Beef boiled in soya sauce	Food items with different brands were purchased from local markets and stored at room temperature.	-	38.7–51.04 9.11–26.58	[17]
98% fat free chicken breast Premium quality corned beef Premium quality Deviled ham spread Corned beef Pork (Spam classic) Chunk white chicken	Three cans of particular foods were collected from local supermarket.	5.70 3.48 2.36 0.78 0.26	1.64–1.73	[80]
Beef Chicken Pork Meat sauce/soup	5 meat and 18 soup or sauces cans were purchased from supermarket.	4	9–10 10–20 11–13	[8]
Canned meat	Single cans of different brands were purchased from major supermarkets.	-	29–98	[9]

* Sd: Standard deviation.

**Table 3 foods-10-00714-t003:** Concentration of BPA found in non-canned/raw meat and meat products.

Figure	Sample Description	BPA Concentration (µg/kg)		Reference
		Mean ± Sd *	Range (Min–Max)	
Beef steak Pork chop Mutton Roast pork Veal	Overall, 20,280 food items were purchased from French territory at regional scale, and prepared as typically consumed by the population.	3.40 ± 6.66 16.95 ± 10.34 7.76 ± 6.43 12.44 ± 17.38 34.41 ± 58.73	0.11–26.91 4.09–40.09 1.71–22.74 2.20–68.92 3.68–223.52	[6]
Beef steak Pork chop Mutton Roast pork Veal	322 non-canned foods of animal origin was collected with two types of packing- pre-packaged and cut-to-order.	2.93 ± 5.51 1.61 ± 2.8 3.19 ± 5.92 3.45 ± 9.04 1.16 ± 1.65	0.09–25.18 0.09–7.03 0.09–18.92 0.09–43.58 0.09–5.72	[16]
Minced meat Chicken fillet Sausages Hamburgers Sliced salami Liver paté Sliced ham Sliced turkey	Food items in plastic packages were collected from grocery store and stored in a refrigerator or a freezer according to specifications written on the label.	0.19 <0.10 2.1 0.17 0.29 3.2 <0.10 0.88	-	[83]
Pork Beef Chicken Mutton Duck	Whole of chicken and duck was purchased, and the skin and internal organs were removed. For pork, thin meat was purchased. Samples were stored at 4 °C until analysis.	0.33 0.73 0.54	0.9–7.08 0.49–0.85	[84]

* Sd: Standard deviation.

**Table 4 foods-10-00714-t004:** Sample preparation and extraction of bisphenols from meat and meat products.

Meat Samples	Type of Bisphenols	Extraction Method	Brief Description of Extraction Method	Reference
Canned chicken	BPA	QuEChERS	Homogenized samples were mixed with acetonitrile, NaCl, MgSO_4_ and extracted with QuEChERS extraction kit, and derivatized.	[77]
Tripe Meat ball Fish and Seafood	BPA, BPB, BPF, BPE, BADGEs	SUPRAS-based microextraction	Solid content of the canned food was homogenized, an aliquot mixed with supramolecular solvent, vortexed, centrifuged. Extract was obtained with glass syringe and used for chromatographic analysis.	[78]
Goulash, caned	BPA	Sol-gel immunoaffinity chromatography	Gel was formed by mixing 1 mL of Phosphate-buffered saline containing 1 mg of BPA antibody with 1 mL of prehydrolyzed tetramethoxysilane. The resulting silica glass was ground in an achate mortar and packed into a 3 mL glass column equipped with a polytetrafluoroethylene frit. Sample was homogenized with acetonitrile and hexane, centrifuged, extracted with acetonitrile, filtered before placed into column, and eluted with acetonitrile/water (40:60, *v*/*v*)	[10]
Luncheon meats, canned Soups, meat, canned	BPA	SPE	Sample mixed with internal standards (BPA-d_16_) were extracted with acetonitrile, cleaned-up through C18 SPE cartridge, and eluted with 50% acetonitrile/water.	[14]
Beef, steak Beef, roast Beef, ground Pork, fresh Veal, cutlets Lamb Luncheon meats, cold cuts Organ meats Wieners and sausages	BPS, BPB, BPAF		Internally spiked samples were mixed with acetonitrile, cleaned-up with Strata-X SPE cartridge, the cartridge was rinsed with 10 mL of 20% acetonitrile in water, and eluted with 10 mL of methanol.	[76]
Meat pates and sausages	BPA, BPB, BPF, BPAF, and BPZ	QuEChERS	Homogenized sample were taken into glass vials containing n-heptane and water. Vials were vortexed after adding acetonitrile, MgSO_4_, and NaCl. An aliquot of the supernatant was added to Z-sep + and C18, mixed and vortexed.	[88]
Meat, poultry and game, offal, delicatessen meats	BPA	SPE	Two successive solid phase extractions (SPE) were performed. The first SPE was carried out using polystyrene-divinyl benzene polymer. After loading the sample, the stationary phase was washed with water, water/methanol (90:10, *v*/*v*) and water/methanol (40:60, *v*/*v*). Analyte elution was done with methanol and load into specific Molecularly Imprinted Polymer (MIP) stationary phase after evaporation and resubmission into acetonitrile. After following the conditioning and washing steps the analytes were eluted with methanol.	[89]
Bovine muscle Cut of bovine meat Roast Pork Raw ham Parma Ham Turkey breast Chicken breast Swine muscle Ovine meat Bovine liver Chipolata sausage	BPA, BPA-G, BPA-2G, BPA-S, BPA-2S	SPE	Sample mixed with internal standards, extracted with water/acetonitrile (50:50) and purified with two successive SPE columns of polystyrene-divinylbenzene polymer and quaternary ammonium SPE SAX cartridge.	[16]
Meat (beef, pork, chicken, duck, sausages)	BPA, BPS, BPF	SPE	Solid samples spiked with internal standards were extracted twice with acetonitrile, purified with NH_2_ cartridges (Strata), and eluted with 80% methanol/acetone.	[15,20]
Beef chicken Pork Meat sauce	BPA	SPE	Homogenized samples were extracted with acetonitrile, passed through solid extraction column (OASIS), eluted with ethyl acetate, dried under N_2_, and dissolved in acetonitrile before analysis.	[8]
Beef Pork Mutton Chicken	BPA	SPE	Sample mixed with celite, ground into powder, packed into a stainless-steel ASE cells containing activated alumina. Acetone was used for the extraction and cleaned-up with amino-propyl SPE cartridge.	[84]
Corned beef, canned	BPA	SLE	Homogenized sample was extracted with acetonitrile, derivatized with acetic anhydride. Sample containing more than 1% fat acetonitrile and trimethylpentane was used.	[9]

QuEChERS: Quick, easy, cheap, effective, rugged, and safe, BPB: Bisphenol B, BPF: Bisphenol F, BPE: Bisphenol E, BADGE: Bisphenol A diglycidyl ether, SUPRAS: Supramolecular solvents, SPE: Solid phase extraction, BPAF: Bisphenol AF, BPZ: Bisphenol Z.

**Table 5 foods-10-00714-t005:** Chromatographic analysis of BPA from meat and meat products.

Chromatographic Analysis	Types of Column (Phase Dimensions (Length × ID; Particle Size) Manufacturer)	Mobile Phase for LC/Carrier Gas for GC	Sensitivity	Linearity and Range	Mean Recovery (%)	Reference
HPLC UV	5 µm Waters C18 column, 250 × 4.6 mm Wakosil 5C18 4.6 mm × 150 mm	Water/acetonitrile (40:60, *v*/*v*); Isocratic conditions 60% Methanol; Isocratic conditions	LOQ: 1.5 mg/kg LOD: 0.8 mg/kg LOD: 25 µg/kg		89.84 89.9	[97,104]
HPLC-FLD	Ultrabase C-18 column (particle size 5 µm, length 250 mm, i.d.4.6 mm) Hypersil ODS C18 column (5 mm, 4.6 × 150 mm)	Water and acetonitrile; Gradient conditions Water and acetonitrile; Gradient conditions	MDL: 0.8 µg/kg MQL: 2.9 µg/kg MQL: 15–113 ng/g	0.9995	80–110 90–99	[78,79]
	C18 column, 150 × 3 mm i.d., 3µm	50 mM sodium acetate buffer (pH 4.8, adjusted with acetic acid) and acetonitrile; Gradient conditions	LOQ: 0.4 to 1.5 ng/mL; LOD: 0.2 to 0.8 ng/mL	0.9993; 0.2–50 ng/ mL	27–103	[10]
HPLC-MS/MS	Waters (1.7 μm, 2.1 mm x 100 mm) attached to a Waters Van Guard BEH phenyl pre-column (1.7 μm, 2.1 × 5 mm).	Water and acetonitrile; Gradient conditions	LOD: 0.18 ng/g	0.99	92.4–102	[76]
	C- 18 column (150 mm × 2.1 mm ID, 3.5 µm)	Methanol and water with 0.1% ammonia; Gradient conditions	LOQ: 1 µg/kg	0.99	91–99	[84]
	Thermo Hypersil Gold column (100×2.1 mm, 1.9 μm)	0.1% formic acid in water (MP A) and 0.1% formic acid in acetonitrile	LOD/LOQ: 0.02/0.06 μg/kg for BPA-G; 0.4/1.2 μg/kg for BPA-2G; 0.09/0.27 μg/kg for BPA-S.			[16]
	Betasil C18 (2.1 × 100 mm, 5 μm) connected to a Javelin guard column (Betasil C18, 2.1 × 20 mm, 5 μm)	Methanol and water; Gradient conditions	LOQ: 0.01–3.14 ng/g	0.99; 0.01–100ng/ml	62–120	[15,20]
	Shim-Pack VP-ODS column (150 × 4.6mm i.d., Shimadzu)	Acetonitrile–water–phosphor c acid (40:60:0.2); Isocratic conditions	0.1 ng/ml (RSD 3.2) for LC-MS; 0.1 ng/ ml (RSD 1.2) for LC-MS/MS	-	71.6–83.9	[8]
	Symmetry C18 (3.5 µm, 150mm × 2.1 mm i.d., Waters)	Acetonitrile/water (40:60), Isocratic conditions	LOD: 0.3 ng/ml	-	93	[99]
GC-MS	Agilent HP-5 ms (30 m × 0.25 mm × 0.25 µm (film thickness)	Helium	LOD: 0.00013 ng/g LOQ: 0.0004 ng/g	0.998	80–99	[77]
	HP-5MS Capillary column (30 m × 0.25 mm × 1.0 µm)	Helium	LOD: 1 ng/g	-	-	[14]
	DB-5MS column (30 m × 0.25 mm I.D. × 0.25 µm film thickness	Helium	LOD: 0.15 µg/kg LOQ: 0.5 µg/kg	0.99; 2.5–200 µg/kg	75–95	[88]
	ZB-5MS (Phenomenex) 30 m × 0.25 mm i.d., 0.25 µm film thickness	Helium	LOD: 0.01 to 0.03 µg/kg LOOQ: 0.03 to 0.08 µg/kg	0.9990; 0–100 µg/kg	100	[89]
	J&W DB5ms, 30m × 0.25mm i.d, 0.25 µm film thickness	Helium	LOQ: 10 µg/kg for <1% fat containing sample; 20 µg/kg for >1% fat containing	-	42–112	[9]

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
