# Peer review of "Bisphenol A and Metabolites in Meat and Meat Products: Occurrence, Toxicity, and Recent Development in Analytical Methods"

_foods, 2021, doi:10.3390/foods10040714_

Round 1
Reviewer 1 Report
The present study fills a gap in the review of studies on the occurrence, toxicity, and recent development in analytical
methods related to BPA and its metabolites in meat and meat products.
The authors carried out a well-structured and detailed review, focusing on the essential aspects in relation to the objectives proposed for carrying out the study.
Figures and tables help the reader to better interpret all the information provided and the writing is clear and comprehensive.
The conclusions are illuminating and are based on everything mentioned throughout the text, answering the initial questions of the study.
Therefore, this manuscript must be accepted for publication after minor changes:
Line 10: The meaning of “BPA” need to be provided.
Line 22: The meaning of “SPE” need to be provided.
Some relevant and recent studies on the field should be discussed:
Almeida, S., Raposo, A., Almeida‐González, M., & Carrascosa, C. (2018). Bisphenol A: Food exposure and impact on human health. Comprehensive reviews in food science and food safety, 17(6), 1503-1517.
Stojanović, B., Radović, L., Natić, D., Dodevska, M., Vraštanović‐Pavičević, G., Balaban, M., ... & Antić, V. (2020). Migration of bisphenol a into food simulants and meat rations during initial time of storage. Packaging Technology and Science, 33(2), 75-82.
Reviewer 2 Report
The review is well structured, very complete and up-to-date.
The correspondence of reference numbers between cited in text and listed in the reference list should be carefully reviewed, as they do not coincide. In particular, from citation in text of reference 7 (Deceuninck et al. [7]) that is listed as reference number 12, the numbers do not match.
Abstract: Please explain (all) the abbreviations (BPA, SPE, etc) the first time they are used in the text
Please consider the transformation of µg kg-1 into µg/kg, and μg (kg bW)-1 d-1 into μg/bw/day and so on
Line 129: The abbreviation TDI usually refers to Tolerable Daily Intake (not tolerable daily unit)
Line 177: please change 0.1 parts per billion to standard concentration units
Lines 260-265: Has BPA or its metabolites studied by IARC?
Line 261: Can you transform the BPA concentrations from mol/L to mass/L?
Table 2: please define spam as meat product
Figure 3: Abbreviations need to be defined
